# Targeted Profiling of Immunological Genes during Norovirus Replication in Human Intestinal Enteroids

**DOI:** 10.3390/v13020155

**Published:** 2021-01-21

**Authors:** Jenny C.M. Chan, Kirran N. Mohammad, Lin-Yao Zhang, Sunny H. Wong, Martin Chi-Wai Chan

**Affiliations:** 1Department of Microbiology, Faculty of Medicine, The Chinese University of Hong Kong, Hong Kong, China; jennyccm@link.cuhk.edu.hk (J.C.M.C.); kirran.mkn@gmail.com (K.N.M.); mikozhang@cuhk.edu.hk (L.-Y.Z.); 2Li Ka Shing Institute of Health Sciences, Faculty of Medicine, The Chinese University of Hong Kong, Hong Kong, China; wonghei@cuhk.edu.hk; 3Department of Medicine and Therapeutics, Institute of Digestive Disease, State Key Laboratory of Digestive Disease, Faculty of Medicine, The Chinese University of Hong Kong, Hong Kong, China

**Keywords:** human norovirus, human intestinal enteroids, innate immune response, virus–host interaction

## Abstract

Norovirus is the leading cause of acute gastroenteritis worldwide. The pathogenesis of norovirus and the induced immune response remain poorly understood due to the lack of a robust virus culture system. The monolayers of two secretor-positive Chinese human intestinal enteroid (HIE) lines were challenged with two norovirus pandemic GII.4 Sydney strains. Norovirus RNA replication in supernatants and cell lysates were quantified by RT-qPCR. RNA expression levels of immune-related genes were profiled using PCR arrays. The secreted protein levels of shortlisted upregulated genes were measured in supernatants using analyte-specific enzyme-linked immunosorbent assay (ELISA). Productive norovirus replications were achieved in three (75%) out of four inoculations. The two most upregulated immune-related genes were CXCL10 (93-folds) and IFI44L (580-folds). Gene expressions of CXCL10 and IFI44L were positively correlated with the level of norovirus RNA replication (CXCL10: Spearman’s r = 0.779, *p* < 0.05; IFI44L: r = 0.881, *p* < 0.01). The higher level of secreted CXCL10 and IFI44L proteins confirmed their elevated gene expression. The two genes have been reported to be upregulated in norovirus volunteer challenges and natural human infections by other viruses. Our data suggested that HIE could mimic the innate immune response elicited in natural norovirus infection and, therefore, could serve as an experimental model for future virus-host interaction and antiviral studies.

## 1. Introduction

Human norovirus (HuNoV) is the leading cause of acute viral gastroenteritis worldwide. Norovirus is currently classified into ten genogroups (GI–GX) based on the sequence variation of the capsid protein (VP1) and the RNA-dependent RNA polymerase and is further subdivided into 48 genotypes (e.g., GI.1 and GII.4) [1]. Norovirus is highly genetically diverse and can infect a wide range of hosts, including humans, mice [2], cows [3,4], pigs [5], dogs [6], cats [7], marine mammals [8] and bats [9]. Among the ten genogroups, five (GI, GII, GIV, GVIII and GIX) are able to infect humans, where GII was reported to be responsible for most of the human infections [10]. Among all the genotypes that are associated with human infection, strains of the GII.4 genotype are the most predominant [11] and have attributed to 70–80% of all reported outbreaks in past decades [12]. Every two to three years, the genotype GII.4 undergoes epochal evolution and new variants often emerge [13]. A total of six immune-escaped variants of the GII.4 genotypes have evolved since 1995 and have resulted in pandemics across the world [14,15], whereas GII.4 Sydney [P31] is the most predominant strain worldwide [16]. In general, all age groups are vulnerable to HuNoV infection, with the majority of severe cases occurring in the elderly and infants. HuNoV infection is generally self-limiting, yet severe cases may occasionally occur in infants, elderly patients, or immunocompromised patients and, in some cases, may lead to chronic infection with persistence gastrointestinal diseases.

While the transmission mode and disease severity are dependent on the HuNoV genotypes, it is also well accepted that the susceptibility of a host is also liable to both virus genotypes and human host factors. A study performed in 2007 by Kindberg and colleagues confirmed that the α(1,2)-fucosyltransferase 2 (FUT2) gene would affect host secretor status, which in turn dictated the host susceptibility towards HuNoV infection [17]. Although host secretor status is determined by functional FUT2 gene expression, the diversity of a FUT2 gene mutation is ethnically dependent.

Millions of HuNoV infections have been reported globally in the past decades, but the pathogenesis of HuNoV and the immune response induced remain poorly understood due to the lack of a robust cell culture system. Numerous efforts had been devoted to the development of an in vitro model for HuNoV cultivation, especially using cell lines derived from human gastrointestinal tract epithelia, but none of them succeeded [18,19,20,21]; researchers also attempted to develop 3D cell culture models by differentiating intestinal epithelial cell line and human epithelial colorectal adenocarcinoma cells in rotating-wall vessels. Although a cytopathic effect and limited viral RNA replication were reported in both attempts [22,23], reproducing these models in other laboratories remained a challenge [24].

Even though acute viral gastroenteritis caused by HuNoV is mild in most cases, it creates a substantial economic burden globally [25]. A better understanding of the pathogenesis and host immune response would help to resolve this burden. However, due to the lack of a robust and reproducible cell culture system and small animal models for HuNoV, an extensive amount of research had been carried out using models that could mimic gastrointestinal diseases. Therefore, most of the previous understanding of HuNoV immune response was from either human volunteer challenge studies [26,27], phylogenetically closely related virus surrogate models such as murine norovirus (MNV) [28,29,30,31], or other larger animals such as calves and pigs that were susceptible to HuNoV presenting major gastroenteritis symptoms [28,32].

The establishment of human intestinal enteroids (HIEs) in 2009 [33] advanced our understanding of the human intestinal epithelium. This advanced stem cell-based technology has also led to a tremendous number of opportunities to study virus pathogenesis and host–microbe interactions in this particular niche. The breakthrough in 2016, where Ettayebi and his colleagues successfully cultivated HuNoV in the HIE model [34], shed light onto a better understanding of HuNoV pathogenesis. Since HIE is the most reproducible culture model to cultivate HuNoV thus far, in this study, we aim to utilise this model to profile the immune response induced under HuNoV infection. In a study that performed staining on duodenal, jejunal and ileal biopsies obtained from HuNoV-positive patients, it was shown that viral protein 1 was detected in different parts of the small intestine, indicating that all segments of the small intestine support HuNoV replication [35]. Since a majority of the studies uses jejunal- and ileal-derived HIE [34,36,37,38], we therefore derived duodenal HIE to investigate the difference between infection sites. Using immune response as a parameter, we further investigated the correlation between immune response and HuNoV replication in HIE and evaluated the robustness of the HIE model by comparing our observations with HuNoV naturally infected cases in the literature and other studies.

Our data showed that, upon successful infection, CXCL10 and IFI44L were induced in a virus-specific manner despite ethnicity and that the level of induction was strongly correlated with the norovirus RNA copy number at the transcriptomic level and was comparable between two virus strains. The enzyme-linked immunosorbent assay (ELISA) results obtained for CXCL10 concentration in supernatants was strongly correlated with viral RNA replications. The two genes were reported to be upregulated in norovirus volunteer challenges and natural human infections by other viruses. Collectively, these data suggested that HIE could mimic the innate immune response elicited in natural norovirus infections and therefore could serve as an experimental model for future virus–host interaction and antiviral studies. This study on immune response may help to further facilitate research on virus–host interactions upon infection, the development of specific treatments, as well as boosts in vaccine efficacy.

## 2. Materials and Methods

### 2.1. Establishment of Human Intestinal Enteroids

Two duodenal biopsy samples were obtained at the proximal horizontal D1 segment from patients during routine endoscopic examination in the endoscopy centre of Prince of Wales Hospital (PWH), Hong Kong. The samples were taken from normal healthy regions which were assessed by the attending doctor who performed the procedure. All subjects gave their informed consent for inclusion before they participated in the study. The study was conducted in accordance with the Declaration of Helsinki, and the protocol was approved by the Ethics Committee of the Joint Chinese University of Hong Kong—New Territories East Cluster Clinical Research Ethics Committee (reference number 2017.130). Human intestinal enteroids were derived immediately through isolation of intestinal crypts from biopsies following a well-established protocol, as previously described [34,39,40]. HIEs were maintained in Complete medium with growth factors (CMGF+) supplemented with 10 μM of a Rho-associated protein kinase inhibitor, refreshed every two days and passaged every 7 days in a ratio of 1:2 to 1:3, as previously published [39].

### 2.2. Host Secretor Status Determination

Saliva samples were collected from the consenting patients for host secretor status determination. The saliva sample was stored at 4 °C until the procedures for HIE establishment were completed and centrifuged at 10,000× *g* for 5 min at room temperature. The supernatants were transferred, boiled at 100 °C for 5 min and aliquoted for storage at −20 °C as of host secretor status phenotyping purpose. The cell pellet remain was stored for genomic DNA (gDNA) extraction followed by host secretor status genotyping.

Histo-blood group antigen phenotyping was performed using an enzyme-linked immunosorbent assay (ELISA) as previously described [41] with slight modification. All experiments were performed three times. The positive threshold of the optical density (OD) readings was calculated using the following formula: 10% above the cut-off, which was defined as OD absorbance at 450 nm of the negative control, plus 0.15. Negative controls, in which primary antibodies were replaced by either diluent only or isotype-matched IgG/IgM, were also included in each experiment.

FUT2 A385T (rs1047781) genotyping was performed using gDNA extracted from the saliva cell pellet, which was thawed at room temperature for gDNA extraction using the QIAmp DNA mini kit following the manufacturer’s instruction. A master mix was prepared with a pre-designed 40× TaqMan^®^ SNP Genotyping Assay (primers + probe mix; Thermo Fisher, assay ID: C___8832449_10) and 2× TaqMan^®^ Genotyping Master Mix (Thermo Fisher, Waltham, MA, USA) and topped up with RNase-free water. FUT2 A385T genotyping was performed using an Applied Biosystems StepOne real-time PCR machine with 2 µL of extracted gDNA as the template.

### 2.3. Conditioned Medium and Reagents for HIE Differentiation

Wnt3a conditioned medium was prepared using the L Wnt-3A cell line (ATCC) cultured with Dulbecco’s Modified Eagle Medium (DMEM)with 10% Fetal bovine serum (FBS) and 0.4 mg/mL of G-418 in a T-75 flask following the manufacturers’ instructions. The media collected were combined and passed through a 0.22-µm filter unit under vacuum, aliquoted into 25 mL per tube and stored at −70 °C for future use.

The 293-Noggin cells were a gift from Dr. Muncan V. Van den Brink GR [42]. Briefly, the cells were grown with DMEM supplemented with 10% FBS and 10 µg/mL of puromycin for 2–3 days until reaching 80% confluency. The cells were trypsinised with Trypsin-Ethylenediaminetetraacetic acid in PBS, neutralised with DMEM with 10% FBS and centrifuged at 200× *g*. The medium was removed, and the cell pellet was resuspended in 20 mL of medium in a T-75 flask and cultured for one week until the medium turned yellow. The medium collected was centrifuged at 1000× *g* for 10 min and filtered through a 0.22-µm filter unit with Surfactant-free cellulose acetate membrane under vacuum, aliquoted into 10 mL per tube and stored at −70 °C for future use.

R-spondin conditioned medium was prepared using the 293T-HA-Rspol-Fc cells (Trevigen, Gaithersburg, MD, USA) following the manufacturers’ instructions. The cells were cultured in DMEM supplemented with 10% of FBS and 300 µg/mL of Zeocin. The medium harvested was filtered through a 0.22-µm filter unit with SFCA membrane under vacuum. The R-spondin conditioned medium was then aliquoted into 10 mL per tube and stored at −70 °C for future use.

Complete medium without growth factors (CMGF-medium) was prepared with advanced DMEM/F12 (Gibco, Waltham, MA, USA) and was supplemented with 100× Glutamax (Gibco), 1 M HEPES buffer (Gibco) and 100 U/mL penicillin-streptomycin (Gibco).

Complete medium with growth factors (CMGF+) was prepared with basal cultural medium CMGF− and was supplemented with 50% of in-house Wnt3A conditioned medium, 20% of in-house R-spondin 1 conditioned medium, 10% of in-house Noggin conditioned medium, 50 ng/mL of EGF (Invitrogen), 10 nM of nicotinamide (Sigma-Aldrich), 10 nM of gastrin I (Sigman-Aldrich, St. Louis, MO, USA), 500 nM of A-83-01 (Tocris, Bristol, UK), 10 µM of SB202190 (Sigma-Aldrich), 1× B27 supplement (Gibco), 1× N2 supplement (Gibco) and 1 mM of n-acetylcysteine (Sigma-Aldrich).

Differentiation medium was prepared with basal cultural medium CMGF− and was supplemented with 5% of in-house Noggin conditioned medium, 50 ng/mL of EGF, 10 nM of gastrin I, 500 nM of A-83-01, 1× B27 supplement, 1× N2 supplement and 1 mM of n-acetylcysteine. 

### 2.4. Stool Filtrates

HuNoV-positive stool samples were chosen from our HuNoV surveillance study. These samples were collected from paediatric and adult patients who were admitted to PWH presenting severe symptoms of acute gastroenteritis and who underwent routine HuNoV diagnosis in the virology lab of PWH, Hong Kong. Samples tested positive for HuNoV were retrieved and subjected to virus genotyping by our team members following the US CDC RT-PCR dual-typing protocol as previously described [43]. Dual typing was performed using a ProFlex™ 96-well PCR System/SimpliAmp™ Thermal Cycler with 5 µL of extracted norovirus RNA as template. Two GII.4 Sydney[P31] strains, CUHK-NS-1127 and CUHK-NS-1190, were selected based on their diagnostic Ct values of 12.9 and 8.0, with the taxonomer data obtained by viral sequencing using Miseq. The 10% stool suspensions were thawed at room temperature for 5 min, centrifuged at 3000× *g* for 5 min and diluted in 10-fold with PBS. The diluted stool suspension was filtered through a 0.22-µm Ultrafree-MC GV centrifugal filter (Merch Millipore) at 12,000× *g* for 4 min. The 1% stool filtrates were then aliquoted and stored at −70 °C until used. Norovirus RNA was extracted from the 1% stool filtrates using the QIAamp viral RNA mini kit (Qiagen) following the manufacturer’s protocol. The norovirus RNA copy number was quantified as previously described [44] using RT-qPCR performed on Applied Biosystems StepOne real-time PCR machine. Absolute viral RNA quantification was performed against a standard curve generated by 10-fold serial dilutions of synthetic HuNoV GII RNA with amount equivalent to 6 × 10^6^, 6 × 10^5^, 6 × 10^4^, 6 × 10^3^, 600, 60 and 6 copies per reaction.

### 2.5. HuNoV Inoculations

Differentiation of 3D enteroids into 2D monolayer was performed as previously described [34]. HuNoV inoculation was performed as biological duplicates using two HuNoV strains with two HIE lines. Approximately 6–8 × 10^4^ of cells were seeded to each of the human placenta collagen-coated 96 wells. Differentiation medium was changed every second day, and the plate was cultured for differentiation for 4–5 days before virus inoculation. Two GII.4 Sydney [P31] strains, CUHK-NS-1127 and CUHK-NS-1190, were used to prepare the inoculum by diluting with an appropriate volume of CMGF- supplemented with 500 μM of Sodium glycochenodeoxycholate (GCDCA). Afterwards, 100 µL of the inoculum was added to the differentiated monolayers at a multiplicity of infection (MOI) of 50 (approximately 3.0 × 10^6^–3.4 × 10^6^ genome equivalents per well). The plates were incubated at 37 °C for 1 h, the inoculum was gently aspirated and the plates were washed with ice-cold CMGF− three times. After the removal of CMGF, 200 µL of differentiation medium with GCDCA was added to each of the wells, and the supernatants were collected at 1, 24 and 72 h post inoculation (hpi). The supernatants were collected in Eppendorf tubes, and cell debris was removed by centrifugation at 3000× *g* for 5 min. Clarified supernatants were collected and divided into 100 µl per aliquots, one for norovirus RNA extraction and one for cytokines measurement, and both aliquots were stored at −70 °C. RLT buffer was added to each of the wells, and lysed cells were collected and stored at −70 °C for cell pellet noroviral RNA extraction.

### 2.6. RT-PCR and qPCR Analysis

Norovirus RNA was double eluted in 40 µL of AVE buffer using the QIAamp Viral RNA mini Kit following the manufacturers’ instructions. Cell pellets stored in RLT buffer were thawed at room temperature, and viral RNA was extracted in 40 µL of RNase-free water using the RNeasy Mini kit (Qiagen, Venlo, The Netherlands). The viral RNA extracted was then quantified as previously described in [44] using RT-qPCR performed on Applied Biosystems StepOne real-time PCR machine. Absolute viral RNA quantification was performed against a standard curve generated by 10-fold serial dilutions of synthetic HuNoV GII RNA with amount equivalent to 6 × 10^6^, 6 × 10^5^, 6 × 10^4^, 6 × 10^3^, 600, 60 and 6 copies per reaction.

### 2.7. Screening of Immune-Related Gene Expression

The selection of pathways was based on the coverage of specific immune genes that were observed in a natural infection. The commercially available “Innate and Adaptive Immune Responses” and “Human Interferon and Receptors” pathway PCR arrays were selected for this study, and each pathway covered 84 immune-related genes. Viral RNA was extracted from cell lysates, as described earlier. All cell pellets collected at all time points were used. Nanodrop was used to measure the concentration and quality of RNA extracted. For each sample, 50 ng of RNA was reverse transcribed into cDNA using the RT^2^ First Strand kit (Qiagen) following the manufacturer’s instructions. The cDNA prepared was then mixed with 2× RT^2^ SYBR Green Mastermix (Qiagen) and RNase-free water, and RT-qPCR was performed on an Applied Biosystems StepOne real-time PCR machine.

### 2.8. HIE Cell Type Markers Detection

Purified RNA was eluted from cell pellets of differentiated HIE monolayers using the RNeasy Mini kit (Qiagen) following manufacturer’ instructions. The concentration of RNA eluted was quantified by Nanodrop (Thermo Fisher). For each sample, 50 ng of RNA was reverse transcribed into cDNA using the SuperScript™ IV VILO™ Master Mix with ezDNase™ Enzyme (Thermo Fisher). Real-time PCR was performed using the TaqMan Fast Advances master mix and TaqMan Gene Expression Assays (Appendix A) in an Applied Biosystems StepOne real-time PCR machine.

### 2.9. Data Analysis

Gene expression was determined based on the ΔΔCt method and was normalized with the Beta-2-Microglobulin (β2M). Ct values of each target gene, housekeeping genes, genomic DNA contamination controls, and negative controls were exported from the qPCR machine and uploaded to the Qiagen’s GeneGlobe Data Analysis Centre for data analysis. Ct values above 35 were considered as undetermined. The Ct values obtained were then normalized with β2M and compared against 1 hpi of each inoculation to obtain the fold change (ΔΔCt). Upregulation and downregulation of the genes were defined by plus or minus 2-fold against data from 1 hpi. A heat map showing fold changes in the gene expressions was generated using the calculated ΔΔCt value based on Ct values by using Prism 8.2.0 (GraphPad).

### 2.10. Measurement of Secreted Protein of Dyregulated Immune Genes (CXCL10 and IFI44L)

All samples were measured with only one freeze-thaw cycle and no more than three whenever practicable. The aliquoted supernatants were thawed on ice for 5 min and were diluted in 10 folds by adding diluents provided by the Human CXCL10/IP-10 Quantikine ELISA Kit (R&D System) and Human Interferon-induced protein 44-like ELISA kit (My Biosource, San Diego, CA, USA). Levels of released CXCL10 and IFI44L were measured utilizing the ELISA kit as per manufacturer’s instructions. ELISA for CXCL10 and IFI44L was performed as three independent experiments.

### 2.11. Statistical Analysis

Correlation of two continuous variables was evaluated by nonparametric Spearman’s rank test by using log-transformed data on Prism 8.2.0 (GraphPad, San Diego, CA, USA). A two-tailed *p* value < 0.05 was considered statistically significant. Multivariate linear regression analysis was performed on PASW Statistics 18 (formerly SPSS Statistics).

## 3. Results

### 3.1. Selection of HIE Lines for HuNoV Inoculation

During the patient recruitment process, seven individuals gave consent to contribute their duodenal biopsy samples to the study. Four were males and three were females, all aged between 52 to 82 years. HIE lines were generated from six of them, with a success rate of 86%. Saliva supernatants obtained from biopsies donor were used in an ELISA-based phenotyping method. The results obtained from phenotyping illustrated that four of the HIE lines (CUHK-ED-2, CUHK-ED-3, ED-5, and ED-7) were secretor positive (Figure 1) for diverse blood groups A, B and AB. While phenotyping, the results obtained above identified the secretor status of the donors. A major concern is that a weak secretor status is commonly found in the Asian population and often produces ambiguous phenotyping data. Therefore, the results were further confirmed by FUT2 A385T genotyping. An SNP allelic discrimination plot was generated after qPCR. Homozygous secretors, heterozygous secretors and homozygous weak secretors were genotyped as AA, AT and TT, respectively. CUHK-ED-2 and ED-7 were homozygous secretors (AA), CUHK-ED-3 and ED-5 were heterozygous secretors (AT), and ED-4 and ED-6 were homozygous weak secretors (TT) (Table 1). Overall, genotyping data agreed completely with phenotyping data. A summary of the characteristics of the HIE lines established in this study is shown in Table 1. Since histo-blood group antigens (HBGAs) in the host are hypothesized to contribute towards host susceptibility to HuNoV infection, considering the secretor status of the HIE line being established, CUHK-ED-2 and CUHK-ED-3 were used for HuNoV inoculation.

### 3.2. Productive RNA Replication of Two GII.4 Sydney [P31] Strains in CUHK-ED-2 and CUHK-ED-3

To investigate the replication efficiency of HuNoV in HIE, two GII.4 Sydney [P31] strains were used to inoculate CUHK-ED-2 and CUHK-ED-3 at an MOI of 50 (approximately 3.0 × 10^6^–3.4 × 10^6^ genome equivalents/well). Both lines were differentiated into 2D monolayers at passage 11. The supernatants and cell pellets were collected separately at 1, 24 and 72 hpi and were quantified using RT-qPCR as described in the Materials and Methods section. A 10-fold increase in RNA amount with reference to that at 1 hpi was used to define successful replication [45]. Viral replication kinetics were shown and summarized in Figure 2 and Table 2. Productive replication of HuNoV was observed in CUHK-ED-2 inoculated with CUHK-NS-1127 and CUHK-NS-1190, with a maximum of 104- and 204-fold increase at 72 hpi in supernatant and cell lysate, respectively. While for CUHK-ED-3 inoculated with CUHK-NS-1127, only marginal replication of 12-fold increase was observed in supernatants collected at 72 hpi, no increase in viral RNA copies was observed in cell pellets. For CUHK-ED-3 inoculated with CUHK-NS-1190, no observable replication of HuNoV was seen in either supernatants or cell pellets. No cytopathic effect was observed in all inoculations at any time point irrespective of virus RNA replication status.

### 3.3. Differentiation Status of HIE 2D Monolayers

Other than quantifying the norovirus replication kinetics in both ED-2 and ED-3 lines, the differentiation status of the 2D monolayers was evaluated by determining the gene expression of six epithelial cells markers (Table 3). RNA extracted from the cell pellets, which were collected from ED-2 and ED-3 inoculated with the two strains, were used and quantified using RT-qPCR. Ct values of intestinal stem cell marker Lgr5 were marginally detected, indicating a low to very low expression of stem cells, where cells in monolayer successfully differentiated. Successful cell differentiation was further confirmed by a high expression of cell markers for enterocytes (SI) and goblet cells (MUC2). For other cell lineage-specific markers, including CHGA, DEFA5 and DCLK1, no gene expression was observed. This suggested that the lack of expression or expression at ultra-low levels of these markers fell below the detection limit of the assay.

### 3.4. Gene Expression of HuNoV Induced Immune-Response

We aimed to utilize RT^2^ PCR arrays to profile elevated immune genes in HIE induced by HuNoV infection using the pandemic GII.4 Sydney [P31] variant. The RNA obtained from cell lysates of inoculated HIE lines was used for cDNA synthesis and gene expression profiling in the two selected immune-related pathways. Considering the three successful inoculations only, CXCL10 and TLR4 were the most up- and downregulated genes in at least one time point (24 and 72 hpi), respectively (Figure 3a) in the “Innate and Adaptive Immune Response” pathway. Upon HuNoV inoculation, a total of 23 and 28 genes were upregulated in CUHK-ED-2 and CUHK-ED-3, respectively. In particular, 14 genes were mutually upregulated in both CUHK-ED-2 and CUHK-ED-3. At the same time, a total of 38 and 15 genes were downregulated in CUHK-ED-2 and CUHK-ED-3, in which 9 genes were downregulated in both CUHK-ED-2 and CUHK-ED-3 (Figure 4). Among them, CXCL10 was the most upregulated gene (range of fold changes: 6.3–93.7). Upregulation of CXCL10 was most evident in HIE line CUHK-ED-2.

A similar analysis was performed using the “Human Interferon and Receptors” pathway PCR array. Considering the three successful inoculations only, IFI44L and CNTFR genes were up- and downregulated in at least one time point (24 and 72 hpi), respectively (Figure 3b). Among the successful inoculations, 42 and 17 genes were upregulated in CUHK-ED-2 and CUHK-ED-3 followed by HuNoV inoculations, where 10 genes were upregulated mutually in both CUHK-ED-2 and CUHK-ED-3. For downregulation, 18 and 28 genes were downregulated in CUHK-ED-2 and CUHK-ED-3, respectively. Notably, 12 of the genes were mutually downregulated in both CUHK-ED-2 and ED 3 (Figure 4). Among them, CXCL10 and IFI44L were the most upregulated genes (range of fold changes: CXCL10 [2.8–519]; IFI44L [3.4–580]). Upregulation of both CXCL10 and IFI44L were most evident in HIE line CUHK-ED-2.

By comparing the two heat maps obtained from the two pathways, CXCL10 was reproducibly upregulated in both pathways with productive HuNoV replication. In addition to CXCL10, IFI44L was the second most upregulated gene among all successful inoculations. Noticeably, upregulation of CXCL10 and IFI44L were observed in both CUHK-ED-2 and CUHK-ED-3 HIE lines (Figure 4), and further analysis was performed on these two genes. On the contrary to upregulation, the fold change obtained from all downregulated genes was only moderate compared to that observed in upregulated genes. Finding an antagonist against upregulated genes was also comparatively easier than finding an antagonist against downregulated genes. As a result, we did not select any downregulated genes for further analysis.

#### Correlation of CXCL10 and IFI44L Gene Expression with HuNoV RNA Replication

The CXCL10 and IFI44L expression fold changes were compared against the cell-associated norovirus RNA fold changes. We observed a strong positive correlation between CXCL10 upregulation and cell-associated norovirus RNA fold change; the higher the upregulation of CXCL10, the higher the cell-associated norovirus RNA fold increases (r = 0.778, *p* = 0.0295; Spearman’s rank test) (Figure 5a). As for IFI44L, upregulation of IFI44L was also observed to be strongly positively correlated with cell-associated norovirus RNA fold increase (r = 0.881, *p* = 0.0072; Spearman’s rank test) (Figure 5b). A multivariate analysis was performed to adjust for other confounders. CXCL10 upregulation and IFI44L upregulation were independently associated with cell-associated norovirus RNA increase after adjusting for HIE lines, virus strains and time points (Appendix A).

### 3.5. Protein Expression of HuNoV-Induced Immune-Response

In order to confirm the transcriptional change in CXCL10 and IFI44L gene expression, the protein expression levels of the two genes were quantified using supernatants collected from the four inoculations by analyte-specific ELISA.

For CXCL10, at 1 and 24 hpi, no noticeable change was seen in all the inoculations. In the supernatants collected from the three successful replications, a significant increase in CXCL10 secretions at 72 hpi was observed, particularly in the CUHK-ED-2 lines, indicating that the infection of HuNoV in HIE could induce CXCL10 secretion, whereas in the unsuccessful case (CUHK-ED-3 inoculated with CUHK-NS-1190), there was no change in the CXCL10 secretion at different time points (Figure 6a).

As for IFI44L, a moderate increase in IFI44L level was generally seen at 24 and 72 hpi in the successful inoculations. No change in the IFI44L secretion was observed in CUHK-ED-3 inoculated with CUHK-NS-1190, resulting in no productive HuNoV replication (Figure 6b).

#### Correlation of CXCL10 and IFI44L Protein Secretion with HuNoV RNA Replication

To examine if the secretion level of CXCL10 and IFI44L was correlated with norovirus RNA replication in HIE, we sought to compare the expression levels of CXCL10 and IFI44L against the norovirus RNA replication in supernatants and cell lysates, respectively. We observed a strong positive correlation between CXCL10 protein secretion and norovirus RNA replication (supernatant: r = 0.8772, *p* = 0.0004; cell pellet: r = 0.7754, *p* = 0.0042; Spearman’s rank test) (Figure 7a,b). To perform a sensitivity test, the correlation test was repeated after the exclusion of the two outliers with very high expression levels (CUHK-ED-2 inoculated with CUHK-NS-1127 and CUHK-NS-1190 at 72 hpi). A positive correlation was found (supernatant: r = 0.7866, *p* = 0.0093; cell pellet: r = 0.6951, *p* = 0.0298; Spearman’s rank test) (Figure 7c,d). A multivariate analysis was performed to adjust for other confounders. CXCL10 secretion was independently associated with supernatant- and cell-associated norovirus RNA increase after adjusting for HIE lines, virus strains and time points (Appendix A).

The IFI44L level in supernatant was also compared against norovirus RNA fold change in both supernatants and cell pellets. However, no correlation was observed (Figure 8), and thus, multivariate analysis was not performed.

## 4. Discussion

Although millions of HuNoV infections have been reported globally, specific treatments and licensed vaccine remain unavailable. The current understanding of pathogenesis and immune response induced by HuNoV is still limited due to the inability in cultivating HuNoV with a robust and reproducible model. However, the breakthrough in 2016, where Ettayebi et al. successfully cultivated HuNoV in HIE, demonstrated a moderate level of norovirus RNA replication within the model [34], and the result was repeatable in many other laboratories. The long-awaited triumph in cultivating this very difficult to culture virus certainly ignited a hope that HIE would become a robust model for HuNoV research.

Here, we have successfully established six enteroid lines from duodenal tissue donated by local patients of either secretors or weak-secretors in our laboratory. Despite the small sample size, we achieved a reasonably high success rate of nearly 90% in creating in-house HIE lines. In order to employ this model to study the susceptibility of HIE lines to HuNoV infection, we first determined the secretor status of the biopsy donors. Saliva samples were collected and used as a surrogate for secretor status determination by phenotyping and genotyping. The results obtained from phenotyping and genotyping concur with each other, which provided essential information on the secretor status of established HIE lines.

We then validated the susceptibility of in-house derived HIE lines using the globally dominating GII.4 Sydney [P31] strains. Based on the current understanding of the importance of host secretor status on HuNoV susceptibility, two enteroid lines from secretors positive for blood groups A and AB, respectively, were chosen in this study. Regarding stool filtrates, two clinical samples with low diagnostic Ct values of around 10 (with high viral loads) were selected. This was to ensure that the stool filtrates would contain a reasonable amount of virus favouring inoculation with a high MOI (to be discussed below). The metagenomic sequencing data obtained demonstrated that HuNoV was the only enteric virus present in the two stool filtrates, eliminating the possibility of co-infection that may affect downstream experiments and, hence, further confirmed the suitability of the stool filtrates selected. Although several studies have mentioned that stool filtrates obtained from children have a higher viral replication rate in HIEs [36,46], the underlying mechanism has yet to be investigated. Considering that the HuNoV burden is highest among infants and elderly, two GII.4 Sydney [P31] strains obtained from a 1-year-old infant and a 79-year-old male were included for validation.

The next parameter requiring optimization was the amount of virus to be added to HIEs. Generally, virus infectivity is measured by plaque assay using a cell line that is susceptible to the virus of interest. However, no immortalized cell lines were susceptible to HuNoV infection, making it impossible to perform a plaque assay to determine the MOI in our case. Thus, we used the viral RNA copy number obtained via RT-qPCR as a surrogate to determine the MOI used for inoculation and to study viral kinetics. In the pilot study, three MOIs were tested: 1 (5.7 × 10^4^ genome equivalents/well), 5 (2.8 × 10^5^ genome equivalents/well) and 50 (approximately 3.0 × 10^6^–3.4 × 10^6^ genome equivalents/well). No robust viral RNA replication was observed at MOIs of 1 and 5 (Appendix A). Consequently, an MOI of 50 was used in all the remaining inoculation experiments. Even at such a high MOI, only a moderate viral RNA replication of about 100–200 folds was obtained at 72 hpi. Our findings were comparable to other groups that reported 10 to 1000-fold increases in viral RNA [34,45,46,47]. However, this level of replication is suboptimal compared to that of other viruses such as influenza viruses, in which millions of folds of genome copy increase in cell culture are often reported [48,49]. This indicates that, although HuNoV is cultivable in HIE, anonymous restrictive factors are present and may influence HuNoV replication in the HIE model. As the stool filtrates were prepared directly from clinical samples with high viral loads, it was presumed that both virus strains were replication-competent. Despite the likelihood in losing virus infectivity during filtrate preparation steps, the high stability of HuNoV to disinfectants and to harsh environment argues against this possibility [46,50]. However, it should be read with caution that viral RNA level does not equal the amount of infectious virus. In a very recent study, by using viability PCR, it was revealed that only around 5% of all norovirus RNA genomes detected by qPCR came from intact virions [51]. Interestingly, one study revealed that HuNoV particles were packed within exosome-derived small vesicles of less than 200 nm [52]. As a deduction, it was proposed that our filtration step using 0.22-µm filters may accidentally filter out these vesicles and reduce HuNoV infectivity. Alternatively, a co-existing mucosal IgA present in stool filtrates may partly neutralize HuNoV replication in HIE as higher faecal IgA has been shown to be protective and to correlate with lower viral shedding in volunteers challenged with the GI.1 Norwalk virus [53]. Collectively, future stool filtrate preparation protocols involving the use of filters with larger pore size and IgA pre-adsorption should be evaluated.

CUHK-NS-1190 (from an infant) appeared to replicate more efficiently than CUHK-NS-1127 (from an elderly), supporting that stool filtrates prepared from infants may replicate better, as reported by other groups [36,46]. In addition, both CUHK-NS-1127 and CUHK-NS-1190 replicated better in CUHK-ED-2 than in CUHK-ED-3, and CUHK-ED-3 was only susceptible to CUHK-NS-1127 but not CUHK-NS-1190, whereas both strains could replicate effectively within CUHK-ED-2. At first, we speculated on whether the variation in differentiation status between CUHK-ED-2 and ED-3 may play a role. We sought to quantify the gene expression of cell markers for representative intestinal epithelial cell types. The results showed a high expression of enterocyte cell markers and barely detectable levels of a Lgr5 cell marker for intestinal stem cells, suggesting that the monolayers were well-differentiated. Nonetheless, the expression level of enteroendocrine cells, Paneth cells and tuft cells were undetectable in our samples. The result may not be surprising if we consider the original ratio of various cell types expressed within the intestinal epithelium [54,55]. Interestingly, studies have revealed that individuals with either blood groups B or AB are less susceptible to certain strains of HuNoV compared to blood group A or O [56,57], inferring that differences in blood groups between CUHK-ED-2 and CUHK-ED-3 may lead to differences in HuNoV replication efficiency.

We did not observe any cytopathic effect amid viral RNA increases in both supernatants and cell pellets. One possible explanation is that HuNoV was released extracellularly without disrupting the plasma membrane via a process known as virus egress that has been reported in other viruses such as influenza virus, Ebola virus and herpesvirus [58,59,60]. The work published by Santiana et al., where HuNoV shed in stools as vesicles of exosomal or membrane origin, further supported this assumption [52]. Moreover, the inability to generate a high titre stock after multiple passages of HuNoV in HIE [34] has raised the concern of whether the replication of HuNoV in HIE would have produced infectious viruses that are capable of doing two-round infection. Due to time and resource limitations, we did not attempt to perform a virus passage in this study.

To investigate the correlation of immune response with norovirus RNA replication in HIE, we utilised RT^2^ PCR arrays and analyte-specific ELISA to study the immune response induced by HuNoV using the pandemic GII.4 Sydney [P31] variant. Due to the lack of a robust cell culture model, most of the current knowledge on immune response to HuNoV infection is founded on research using animal models. Many studies reported the importance of type I and III interferon responses to norovirus infection in mice, in which they emphasized that the stimulation of innate immunity may mediate more efficient protection against norovirus infection [2,61]. Baldridge et al. also proposed that the activation of innate immune responses may contribute to persistent enteric norovirus infection [62]. In addition to MNV studies, the innate immune response induced by HuNoV has also been studied using gnotobiotic pigs and calves. Upon infection, IFN-α as well as pro-inflammatory cytokines involved in Th1 and Th2 responses were induced [28,29,32]. Therefore, both innate immunity and IFNs were the most well-studied responses based on animal models, which helped in selecting appropriate PCR array pathways. Two recent studies using RNA-seq on HIE have shown that interferon responses were mounted upon human GII.4 norovirus infection [37,38], which has strengthened our choice of PCR array pathways.

At last, two pathways, “Innate and Adaptive Immune Response” and “Human Interferon and Receptors”, were chosen to be investigated using our samples with the RT^2^ PCR array. After selecting appropriate PCR array pathways, we then investigated the transcriptional immune response induced by HuNoV infection in HIEs using PCR arrays. Samples collected from the four inoculations using two GII.4 Sydney [P31] strains on two enteroid lines were used. Three out of four inoculations gave productive HuNoV RNA replications. Instead of being a drawback, the unsuccessful case, CUHK-ED-3 inoculated with CUHK-NS-1190, was used as a negative control to monitor the change in immune genes compared to the three successful cases.

In HuNoV inoculated enteroids, we detected a robust induction of CXCL10 in both arrays, indicating that an antiviral response was triggered in response to HuNoV infection. We found that the gene expression of CXCL10 was most upregulated and was reproducible in both HIE lines inoculated with two separate virus strains. On the contrary, from the unsuccessful inoculation (CUHK-ED-3 inoculated with CUHK-NS-1190), no upregulation was detected. Instead, CXCL10 was surprisingly downregulated by 10 and 8 folds at 24 and 72 hpi, respectively. This may represent a background CXCL10 response in the experimental settings, and the actual CXCL10 upregulation in response to HuNoV replication may therefore be even higher. CXCL10, also commonly known as IP-10, is a chemokine induced by IFN-γ first reported in 1985 [63]. The upregulation of CXCL10 was an innate response induced by viral infections. The upregulation of CXCL10 was also observed in two HuNoV naturally infected cases. In a clinical study of hospitalized HuNoV gastroenteritis patients, an elevated serum level of CXCL10 was observed [64]. Another study also reported an early increase of IP-10 in a patient 2 days after HuNoV infection [65]. In other surrogate models, a study utilizing microarray, qPCR and ELISA performed on a RAW264.7 macrophage cell line infected with murine norovirus 1 also demonstrated the upregulation of CXCL10 [66]. A similar observation was reported in other viruses. For example, enterovirus 71 demonstrated the induction of such a response upon infection in enteroids [67]. Collectively, our data are in line with the current understanding on CXCL10 in viral infections.

We next investigated the relationship between the upregulated gene expression of CXCL10 with HuNoV RNA replication. Using a statistical correlation test and multivariate analysis, we have shown that there is a strong positive correlation between the norovirus RNA replication fold change and the CXCL10 expression fold change. This suggested that the change in norovirus replication was likely to contribute towards the change in expression of CXCL10. On the other hand, we observed that the replication of HuNoV usually peaked at 24 hpi and then declined at 72 hpi in both supernatants and cell lysates. Considering that the HuNoV replicated the best at 24 hpi and that the CXCL10 expression fold change peaked at 24 hpi, our data suggested that the replication of HuNoV induced a potent innate immune response, which may help to suppress the viral replication at an early stage.

In order to prove whether the replication of HuNoV induced a potent innate immune response at the protein level, we utilized a commercially available CXCL10 ELISA kit to measure the protein expression level of this gene in supernatants. From the ELISA data, we can confirm that the observed transcriptional changes in CXCL10 correlate with the production of CXCL10 in supernatants. Interestingly, the gene expression of CXCL10 peaked at 24 hpi, whereas the concentration of CXCL10 secreted in supernatants peaked at 72 hpi. This delayed protein expression is, indeed, a piece of strong evidence supporting the induction of a strong early innate immune response, which may help to suppress the viral replication at 72 hpi. Further experiments are required to prove whether the secretion of the CXCL10 protein may help to suppress viral replication.

In addition to CXCL10, we also observed a robust induction of IFI44L. IFI44L is a paralog gene of IFI44, which functions by binding to the protein FKB5 that interacts with kinase and is involved in the induction of type I and type III IFN signalling [68]. During viral infection, the upregulation of IFI44L is commonly seen. A study revealed the potential use of IFI44L as a pan-viral marker to distinguish viral infection from the bacterial counterpart in humans [69]. The induction of IFI44L was shown to have a modest antiviral activity towards hepatitis C virus [70], rotavirus [40] and several enteroviruses [67]. Although little is known about the role of IFI44L in HuNoV infection, considering rotavirus being a gastrointestinal virus that induces similar clinical symptoms like HuNoV, we may use its effect on human rotavirus infection as a reference. In a study conducted by Saxena et al., the induction of IFN in intestinal epithelial cells under human rotavirus infection did not restrict rotavirus replication. In our study, although the transcriptional change in IFI44L gene was shown to be positively correlated with norovirus RNA replication, the secreted protein levels of IFI44L did not correlate with norovirus RNA replication in both supernatants and cell lysates, indicating that the induction of IFI44L may not restrict the replication of HuNoV. One possible explanation is that there are different mechanisms in gene regulation, especially on posttranscriptional and posttranslational regulation, where the changes in transcriptional level to protein expression level varies between genes while protein degradation within the sample may also affect the accuracy in measurement. A couple of studies suggested that some IFN production may facilitate cellular homeostasis by acting as a negative regulator for excessive innate immune response [71,72]. Recently, a study revealed a novel function of IFI44L as a negative regulator that would negatively regulate the induction of type I and III IFN [68], which may explain why the induction of IFI44L may not necessarily restrict HuNoV replication.

We also measured the secretion level of IFI44L in supernatants using a commercially available kit and observed an elevated level generally at 72 hpi. However, the increase at 72 hpi had no association with HuNoV replication in a statistical correlation test. Technically, the sensitivity of the ELISA kit used for measuring IFI44L was much lower than that of CXCL10 (0.05 ng/mL versus 4.5 pg/mL). This may have limited our ability to detect a noticeable change in the secretion of IFI44L. Also, the volume of supernatants available for secreted protein measurement was very scarce and that was the reason why we only selected two upregulated candidate genes for further analysis on protein levels in this study. The use of multiplex immunoassays such as Bio-Plex (Biorad) may be considered in the future to overcome the sample availability problem.

Like all studies, our study has several limitations that are worthy of attention. Firstly, the sample size is small to provide better insight. Due to a limited supply of HIEs, we only inoculated two secretor enteroid lines with two stool filtrates and only included biological replicates (two enteroid line with same HuNoV inoculum) but not technical replicates in our study. Secondly, we only chose two pathways to study and compare the immune response induced in HIE by HuNoV. Ideally, a genome-wide transcriptomic approach, such as RNA-seq, should be employed. However, surprisingly, our results were highly comparable with a genome-wide study that also identified CXCL10 and IFI44L as top five upregulated genes in HIEs under HuNoV infection [38]. In this UK study, the authors tested HuNoV in duodenal and terminal ileal organoids derived from ethnic Caucasians and performed RNA-seq in terminal ileal organoids only. Another study that performed transcriptomic analyses also reported a statistically significant increase in the expression of IFI44L at 24 hpi in Caucasians jejunal enteroids inoculated with GII.4 norovirus [37]. By comparing our data with the two studies, it likely suggested that HuNoV may induce a general immune response in HIEs irrespective of its position within the gastrointestinal tract and host ethnicity. Our work extends the current literature that the immune response induced upon HuNoV infection in HIE can be generalized across ethnic groups, intestinal anatomical sites and virus strains.

In summary, we validated the susceptibility of two GII.4 Sydney [P31] strains in two of our secretor-positive duodenal HIE lines of Chinese ethnicity. Following inoculation, we investigated the immune response induced at both the transcriptional and translational levels, and the results were compared with those observed in either clinical cases, controlled human infection models and animal models in the literature. Our data demonstrated that HIE would respond to HuNoV infection by inducing antiviral signalling pathways, including both innate immune response and interferon responses. This suggests that HIE could mimic the innate immune response elicited in natural HuNoV infection without co-culturing with immune cells, and these signals are induced in a virus-specific manner despite of ethnicity. Therefore, HIE could serve as a good experimental model for virus–host interaction and antiviral studies. The identification of these upregulated genes may help to further investigate the role of innate immune and interferon response upon HuNoV infection and, eventually, aid the development of drugs and vaccines to target HuNoV infection. These immune pathways may also be exploited to further improve the HIE model for HuNoV culturing.

## Figures and Tables

**Figure 1 viruses-13-00155-f001:**
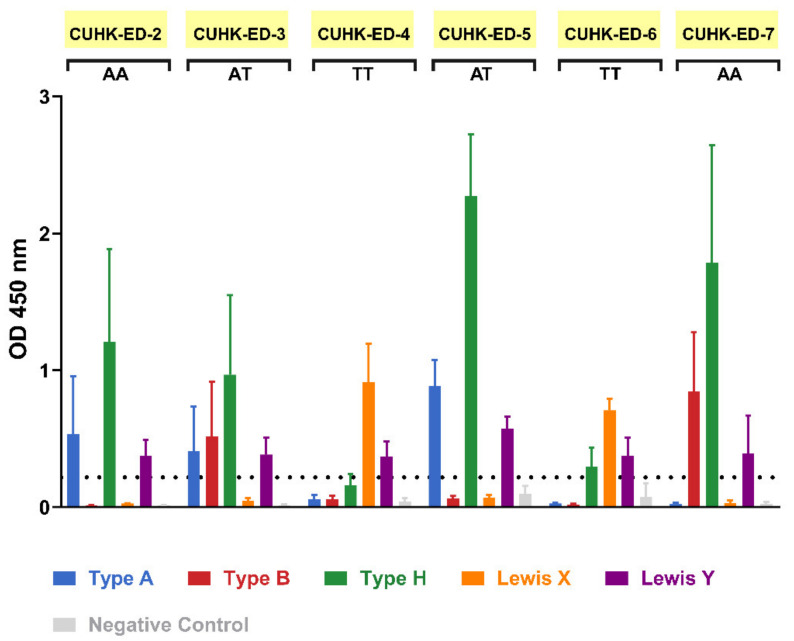
Secretor status phenotyping of enteroid donors obtained via an enzyme-linked immunosorbent assay (ELISA)-based method: saliva phenotyping was performed as three independent experiments. FUT2 A385T genotype status is shown below each sample name, where AA and AT refer to secretor positive and TT refers to weak secretor. For the negative control, HBGA-specific primary antibodies were replaced with one of the followings: diluent (for CUHK-ED-2, CUHK-ED-3 and ED-4), IgG isotype control (for ED-5) and IgM isotype control (for ED-6 and onwards). The horizontal dotted line represents an optical density (OD) absorbance of 0.2 at 450 nm. Error bars represent the standard deviation of the mean of the three technical replicates.

**Figure 2 viruses-13-00155-f002:**
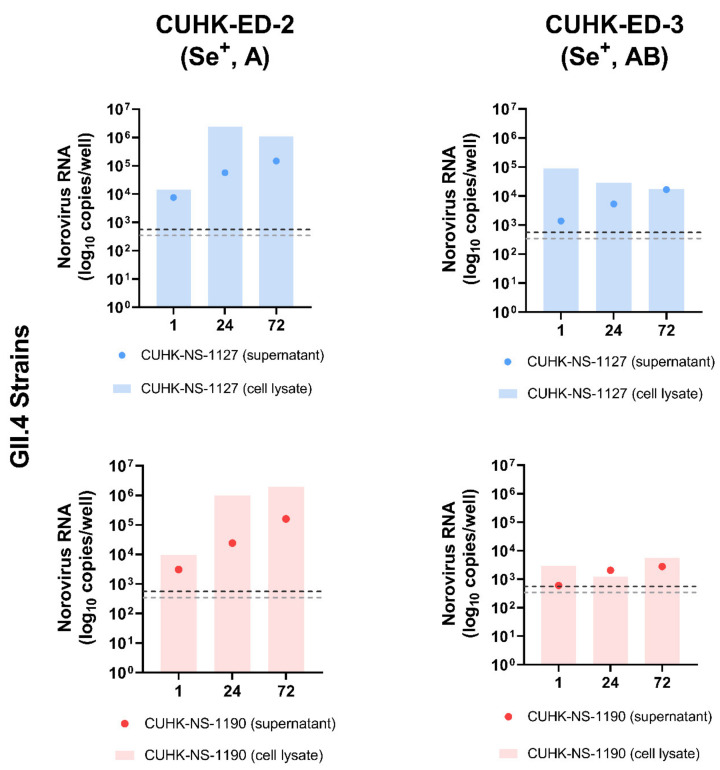
Replication of two Human norovirus (HuNoV) GII.4 Sydney [P31] strains in human intestinal enteroids (HIE) lines CUHK-ED-2 and CUHK-ED-3: The monolayer of each human intestinal enteroids lines was inoculated at a multiplicity of infection of 50 (approximately 3.0 × 10^6^–3.4 × 10^6^ genome equivalents/well) with the two strains in two independent experiments (as biological replicates). Norovirus RNA was extracted from the supernatant and cell lysates collected at 1, 24 and 72 h post-inoculation (hpi). Viral RNA copies per well were obtained by a highly sensitive genogroup-specific quantitative reverse transcription polymerase chain reaction (RT-qPCR). The horizontal grey and silver dotted lines indicate the lower detection limit of RT-qPCR at 552 copies per well of supernatant and 342 copies per well of cell lysates, respectively. Se^+^, indicates secretor positive; A and AB, represent blood groups A and AB, respectively.

**Figure 3 viruses-13-00155-f003:**
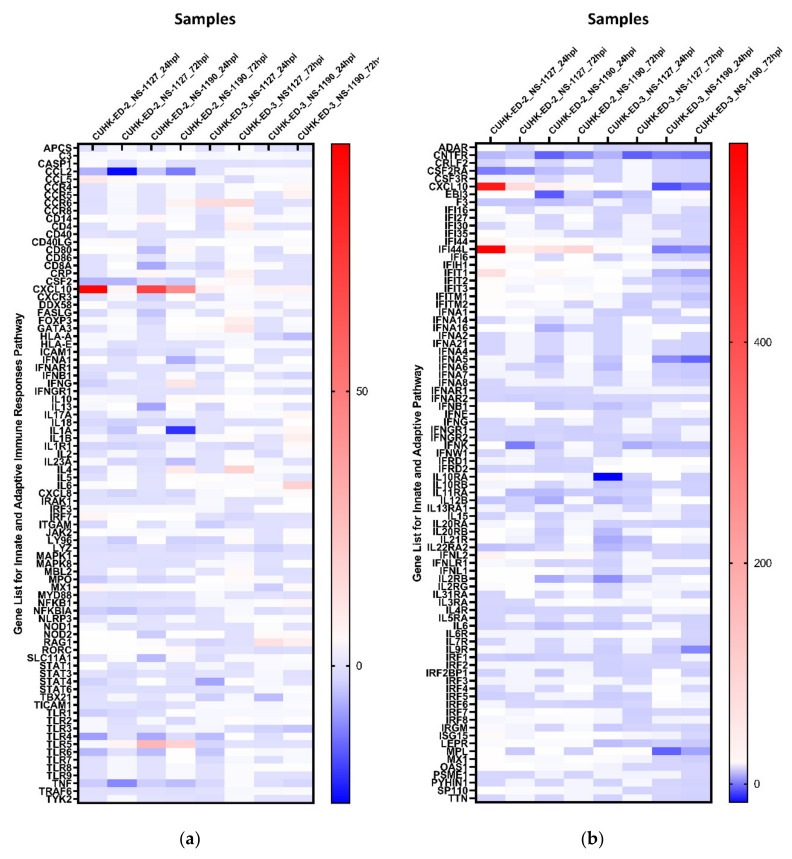
Heat maps showing the regulation of genes expression within each inoculation against 1 hpi of (**a**) the “Innate and Adaptive Immunity” pathway and (**b**) the “Human Interferon and Receptor” pathway: a gradient of blue to red was used to indicate the fold change obtained. Red colour illustrates an upregulation of the genes, whereas blue colour represents a downregulation of genes with reference to baseline samples at 1 hpi. All samples are associated with productive HuNoV replication except CUHK-ED-3 inoculated with CUHK-NS-1190.

**Figure 4 viruses-13-00155-f004:**
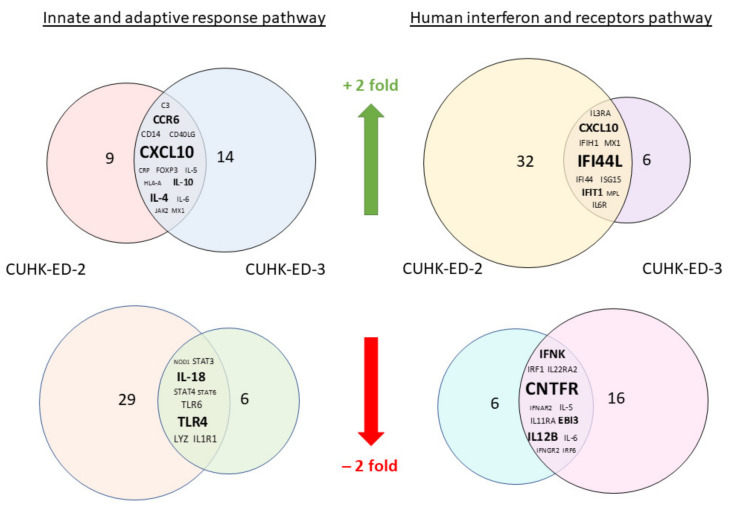
A Venn diagram showing genes that were either mutually upregulated or downregulated between CUHK-ED-2 and CUHK-ED-3 in the two selected pathways: the mutually upregulated and downregulated genes are listed in the overlapped area between the two circles. The font size of the genes is relatively proportional to the fold change of gene expression.

**Figure 5 viruses-13-00155-f005:**
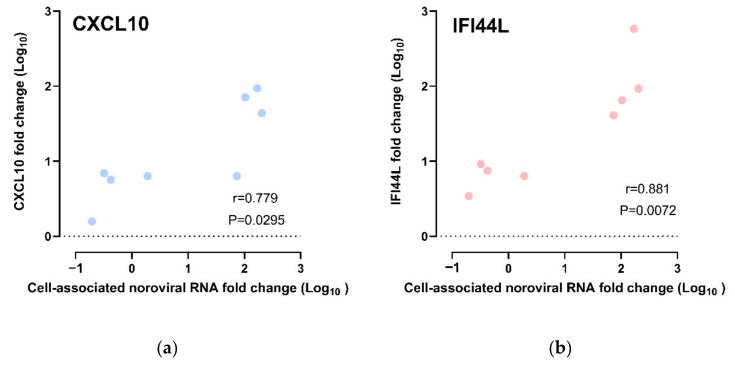
Correlation of gene expression fold change and cell-associated norovirus RNA fold change: All data points were included, including those from CUHK-ED-3 inoculated with CUHK-NS-1190 that showed no productive HuNoV replication, for (**a**) CXCL10 and (**b**) IFI44L.

**Figure 6 viruses-13-00155-f006:**
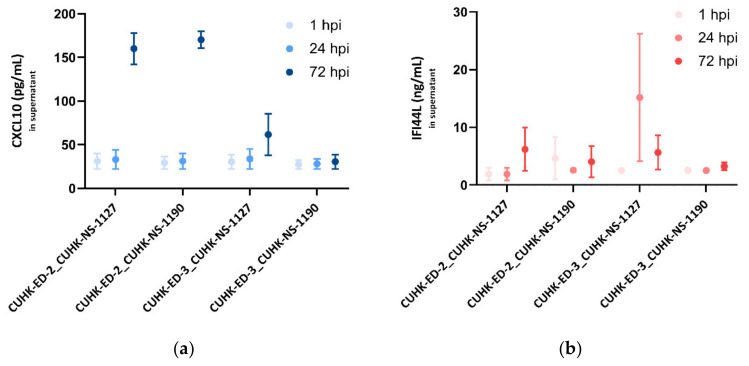
Secreted (**a**) CXCL10 (pg/mL) and (**b**) IFI44L (ng/mL) levels in enteroid lines at 1, 24 and 72 hpi as measured by ELISA: three independent experiments as technical replicates were performed using the supernatants infected with CUHK-NS-1127 and CUHK-NS-1190. Error bars denote standard error of the mean.

**Figure 7 viruses-13-00155-f007:**
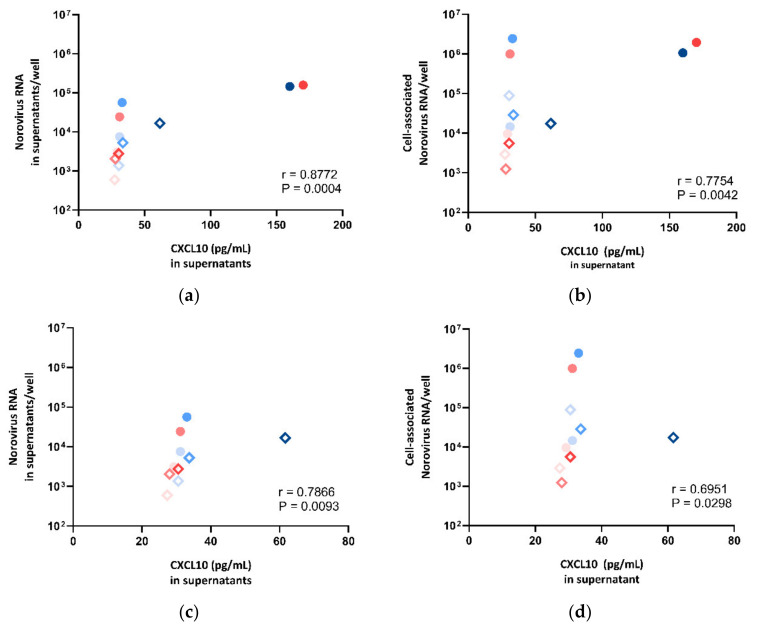
Correlation between norovirus RNA in (**a**) cell lysates, (**b**) supernatants, (**c**) cell lysates after dropping two outliers and (**d**) supernatants after dropping two outliers against CXCL10 concentration measured in supernatants, where the symbol shape indicates the enteroid line used: circles represent CUHK-ED-2, and hollow rhombuses represent CUHK-ED-3. The colour indicates the virus strain for inoculation: blue was used for CUHK-NS-1127 and pink was used for CUHK-NS-1190. The gradient of colour change illustrates the time of sampling: the lightest colour denotes 1 h post-inoculation (hpi), the colour in between denotes 24 hpi and the darkest colour indicates 72 hpi.

**Figure 8 viruses-13-00155-f008:**
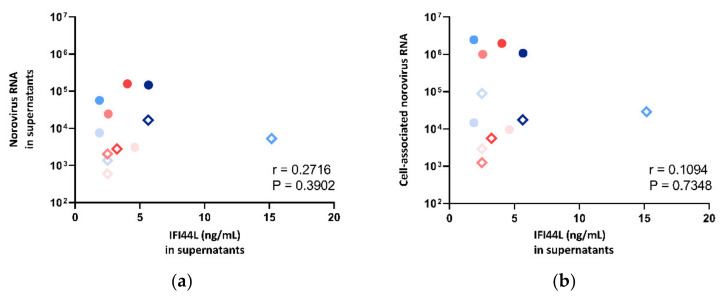
Correlation between norovirus RNA in (**a**) cell lysates and (**b**) supernatants against IFI44L concentration measured in supernatants, where the symbol shape indicates the enteroid line used: circles represent CUHK-ED-2, and hollow rhombuses represent CUHK-ED-3. The colour indicates the virus strain for inoculation: blue was used for CUHK-NS-1127, and pink was used for CUHK-NS-1190. The gradient of colour change illustrates the time of sampling: the lightest colour denotes 1 h post-inoculation (hpi), the colour in between denotes 24 hpi and the darkest colour indicates 72 hpi.

**Table 1 viruses-13-00155-t001:** Demographic characteristics of donors and secretor status of human intestinal enteroid (HIE) lines established in this study.

HIE Lines	Gender/Age	Site	HBGA Expression	Blood Group	FUT2 A385T
CUHK-ED-1	F/-	D1	-	-	-
CUHK-ED-2	M/82y	D1	Secretor	A	AA
CUHK-ED-3	F/52y	D1	Secretor	AB	AT
CUHK-ED-4	M/62y	D1	Weak	-	TT
CUHK-ED-5	F/55y	D1	Secretor	A	AT
CUHK-ED-6	M/77y	D1	Weak	-	TT
CUHK-ED-7	M/57y	D1	Secretor	B	AA

**Table 2 viruses-13-00155-t002:** Fold change of norovirus RNA increase in supernatants and cell lysates at 24 and 72 hpi against 1 hpi in four inoculations.

Virus Strain	Time Point (hpi)	CUHK-ED-2	CUHK-ED-3
Supernatant: CUHK-NS-1127	24	7.50	3.85
	72	19.40	12.19
Cell lysate: CUHK-NS-1127	24	168.44	0.32
	72	73.31	0.20
Supernatant: CUHK-NS-1190	24	7.90	3.43
	72	104.24	0.42
Cell lysate: CUHK-NS-1190	24	51.10	4.63
	72	204.23	1.91

**Table 3 viruses-13-00155-t003:** Relative gene expression level of intestinal epithelial cell markers in differentiated HIE 2D monolayers using cycle threshold (Ct) value of RT-PCR assay as a proxy.

Cell Marker	Cell Type	Ct Value (Range)
LGR5	Stem cells	35.4–41.0
SI	Enterocytes	22.1–25.5
MUC2	Goblet cells	26.0–31.2
CHGA	Enteroendocrine cells	Undetermined
DEFA5	Paneth cells	Undetermined
DCLK1	Tuft cells	Undetermined

## Data Availability

Data available on request due to a suitable publicly accessible repository is not available.

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
