# Peer review of "Targeted Profiling of Immunological Genes during Norovirus Replication in Human Intestinal Enteroids"

_viruses, 2021, doi:10.3390/v13020155_

Round 1
Reviewer 1 Report
Chan et al. have shown that HIE lines infected with HuNoV induced robust expression CXCL10 and IFI44L at the RNA and protein level. These changes were in concert with increased replication of HuNoV.
The authors suggest that these findings provide a model to study virus-host interactions that is more akin to human infections. I applaud the authors on getting this time-consuming and technical system set-up in their lab, however the transcriptomic and proteomic analysis is completely superficial. The authors fail to characterise there genes of interest in any detail. Rather, they simply catalogue expression levels which alone is not informative for understanding virus-host interactions. The study simply lacks rigour to consider that these two genes play a role in NoV biology.
Furthermore, the authors are not the first group to show transcriptiomic changes as a a result of HuNoV infection in HIE models. Indeed this has already been performed more comprehensively using RNA-sequencing by Lin et al. 2019 (Am J Trans Res), Lin et al. 2020 (PNAS) amongst others. Its not clear at all how this work differs from what is already published in the field or how the data presented in this manuscript offers additional novelty. Have the authors considered whether their work aligns well with what others in the field have published? Although the findings are interesting, they are at best incremental.
Reviewer 2 Report
Chan et al describe generating and evaluating HIE from several different donors with a different HBGA profile and describe successful (~100-fold increase viral RNA copies) replication. They then use commercial kits to test supernatant and cells after infection for an increase/decrease in expression of genes involved in the innate immune response. The found that CXCL10 (93-fold) and IFI44L (580-fold) were the two most up-regulated genes and confirmed this by measuring protein expression. The title of the paper suggests that the majority of the presented research describing immune responses against human norovirus which obviously is not possible as the HIE does not include any cells from the gut-associated lymphoid tissue. In addition, the majority of this paper describes how the HIE from the 6 donors were selected and tested which is really the core of this paper and by itself could stand-alone without the additional information of the upregulated genes as these data don’t answer a specific hypothesis.
Minor comments: Suggest sticking to norovirus instead of noroviral. The paper can be condensed by not repeating information in the Discussion that is also described in the Results and/or M&M sections. In addition, the introduction should be an introduction to make the content of the paper understand for readers so some information (e.g., 57-74) that is not relevant can be deleted without impacting the message of this manuscript. The last paragraph of the Introduction gives already away the conclusion of the paper which is typically information that belongs in the first paragraph of the Discussion.
Reviewer 3 Report
The manuscript from Chan et al. evaluated the reproducibility of growing human
noroviruses (HuNoVs) in a human intestinal enteroids (HIEs) monolayer culture. The
authors showed the viral replication of two GII.4 Sydney strains in two secretor-positive duodenal HIEs. Viral replication was quantified in culture supernatants and in cell lysates after 24 and 72 hpi. They also investigated the immune response in HIE cultures to norovirus infections using PCR arrays and analyte-specific ELISA. They reported gene expression of two up-regulated immune-related genes, CXCL10 and IFI44L, correlates to the level of viral replication. The manuscript is well written and the data presented confirm the reproducibility of the HuNoV culture system and the previous findings. I would recommend the acceptance of the paper if the authors address the following points, which would greatly strengthen the manuscript.
The specific comments are as follows:
- The use of PCR arrays and AS ELISA deciphers the up-regulation of two genes, CXCL10 and IFI44L. The paper would be strengthened by including references from other groups. The manuscript needs to acknowledge the previous findings. In fact, up-regulation of CXL10 and IFI44L in norovirus infected HIEs was previously reported by Hosmillo et al (mBio, 2020) and Lin SC. et al (PNAS, 2020).
- Please add titers and TCID50 of each viral sample.
- It would be convenient to use genome equivalents/well for inoculum instead of using MOI. If the authors prefer to include MOI, it would be recommended to show how MOI was calculated.
- Line 39: Reference 2 (Karst et al) talked about MNV that grows in mice and not in rats. Please change. For rat norovirus, the authors can cite Moreira et al (Exp Anim, 2019).
- Line 107: please change to 1 μM of ROCK inhibitor.
- Line 132: The media collected
- Line 181: Specify which collagen used.
- Include primers used for FUT2 genotyping.
- What is the explanation for no/slight replication in ED-3. Did the authors try to infect in presence of bile and see if they can enhance infection of ED-3 HIE line?
- Line 490: The authors list “data not shown” regarding the proliferation/differentiation markers. Since this is an important point for others in the field, this data should be shown in a figure.
- Line 521: Authors stated immune responses in mice and pigs. Please add work done in HIEs [Hosmillo et al (mBio, 2020) and Lin et al (2020)].
Round 2
Reviewer 1 Report
Nil.
